# DE-YOLOv13-S: Research on a Biomimetic Vision-Based Model for Yield Detection of Yunnan Large-Leaf Tea Trees

**DOI:** 10.3390/biomimetics10110724

**Published:** 2025-10-30

**Authors:** Shihao Zhang, Xiaoxue Guo, Meng Tan, Chunhua Yang, Zejun Wang, Gongming Li, Baijuan Wang

**Affiliations:** 1School of Mechanical and Electrical Engineering, Wuhan Donghu College, Wuhan 430071, China; zhangshihao@wdu.edu.cn (S.Z.); 18488871225@163.com (X.G.); t1793630053@163.com (M.T.); 2College of Tea Science, Yunnan Agricultural University, Kunming 650201, China; yang_960204@163.com (C.Y.); wangzejun0529741x@163.com (Z.W.); 3Yunnan Organic Tea Industry Intelligent Engineering Research Center, Yunnan Agricultural University, Kunming 650201, China

**Keywords:** visual mechanisms, DE-YOLOv13-S, dynamicconv, efficient mixed-pooling channel attention, scale-based dynamic loss

## Abstract

To address the challenges of variable target scale, complex background, blurred image, and serious occlusion in the yield detection of Yunnan large-leaf tea tree, this study proposes a deep learning network DE-YOLOv13-S that integrates the visual mechanism of primates. DynamicConv was used to optimize the dynamic adjustment process of the effective receptive field and channel the gain of the primate visual system. Efficient Mixed-pooling Channel Attention was introduced to simulate the observation strategy of ‘global gain control and selective integration parallel’ of the primate visual system. Scale-based Dynamic Loss was used to simulate the foveation mechanism of primates, which significantly improved the positioning accuracy and robustness of Yunnan large-leaf tea tree yield detection. The results show that the Box Loss, Cls Loss, and DFL Loss of the DE-YOLOv13-S network decreased by 18.75%, 3.70%, and 2.54% on the training set, and by 18.48%, 14.29%, and 7.46% on the test set, respectively. Compared with YOLOv13, its parameters and gradients are only increased by 2.06 M, while the computational complexity is reduced by 0.2 G FLOPs, precision, recall, and mAP are increased by 3.78%, 2.04% and 3.35%, respectively. The improved DE-YOLOv13-S network not only provides an efficient and stable yield detection solution for the intelligent management level and high-quality development of tea gardens, but also provides a solid technical support for the deep integration of bionic vision and agricultural remote sensing.

## 1. Introduction

As a distinctive pillar industry in Yunnan Province, the tea sector not only drives local economic development and increases farmers’ income but also has far-reaching impacts on rural revitalization, ecological conservation, and cultural heritage. Within this context, Yunnan large leaf tea trees, as one of the core germplasm resources, account for more than 90 percent of the planting area, and their yield level is directly linked to the stability and high-quality development of the tea industry chain [1]. Variations in tea tree yield are often influenced by physiological traits such as photosynthetic efficiency, nutrient uptake capacity, and resistance to pests and diseases. These traits can, to some extent, serve as key indicators for assessing plant health status [2,3,4]. Therefore, achieving the efficient and accurate monitoring of the yield of Yunnan large leaf tea trees has become a key issue that must be addressed to advance intelligent management of tea plantations [5].

At present, yield monitoring in tea plantations relies primarily on manual estimation and sample surveys, which are inefficient and labor intensive, and are vulnerable to human judgment bias and environmental disturbances that undermine the accuracy and stability of the results [6]. To address this issue, bionic vision technology is emerging as a key pathway for advancing the intelligent transformation of the tea industry [7]. By simulating the overhead perspective and perceptual mechanisms of flying animals such as birds [8], unmanned aerial vehicle (UAV) remote sensing systems can rapidly acquire key data on canopy structure, leaf count, and spatial distribution patterns. This enables efficient and precise monitoring and analysis of yield in Yunnan large-leaf tea trees.

Inspired by the human visual system, Xiaoge Wang et al. proposed an improved small-object detection algorithm, YOLO-HVS, to tackle target recognition in complex occlusion and low signal-to-noise ratio infrared imaging [9]. Experimental results show that YOLO-HVS achieves an mAP (mean average precision) of 83.4% on the DroneVehicle dataset and 97.8% on the DroneRoadVehicles dataset, delivering gains of 1.1% and 0.7% over YOLOv8, respectively, and providing a practical solution for unmanned aerial vehicle remote sensing detection.

To overcome the limitations of traditional tea yield estimation methods, ShuMao Wang et al. introduced an approach that integrates UAV remote sensing with deep learning [10]. Using YOLOv5 as the base network and upgrading the backbone with CSPDarknet53, the method achieved 85.63% mAP, 84.91% precision, 72.19% recall, and 78.00% F1 score. With the introduction of the Squeeze and Excitation module and dual non-maximum suppression, redundant bounding boxes were markedly reduced and model accuracy was improved, providing an intelligent and automated solution for conventional tea yield estimation.

To address the insufficient accuracy of detecting tea diseases in UAV remote sensing imagery, Yongcheng Jiang and colleagues proposed SDDA-YOLO based on the YOLOv8 algorithm [11]. In the image preprocessing stage, the SERB-Swin2sr network is used to reconstruct UAV imagery. In the detection stage, the Shuffle Dual-Dimensional Attention module is introduced to strengthen feature fusion, and the Xsmall-scale detection layer improves the detection of small lesions. The results demonstrate that SDDA-YOLO improves the detection accuracy and mAP by 4.2% and 1.6%, respectively, providing an effective approach to the challenges of accuracy and efficiency in disease detection from UAV imagery.

To resolve issues of low image resolution, severe target occlusion, and limited feature extraction in disease detection for Yunnan large-leaf tea trees, our research team proposes an efficient deep learning-based method integrated with UAV remote sensing [12]. Building on the YOLOv10 network, the method introduces Shape-IoU, wavelet transform convolution, and a Histogram Transformer to deeply optimize the model target localization mechanism and feature representation capacity. Experimental results show that the optimized network achieves increases of 3.4 percent in precision, 10.05 percent in recall, and 11.95 percent in mAP on the test dataset. This study provides an efficient and intelligent tool for agricultural disease detection and indirectly confirms the important value of bionic vision in artificial intelligence.

These studies have demonstrated the considerable advantages of deep learning in target detection, image reconstruction, and multi-scale modeling, thereby providing practical and scalable technical support for crop detection in complex agricultural scenarios [13]. However, experimental results indicate that the aforementioned research still exhibits significant limitations when applied to yield detection tasks in large-leaf tea trees in Yunnan. Constrained by the static architectural configuration of convolutional networks, most current deep learning models struggle to effectively address challenges such as variations in target scale, complex textural characteristics, and severe occlusion prevalent in complex tea garden environments.

To address these limitations, this study proposes DE-YOLOv13-S, a novel deep learning network based on YOLOv13 [14], that integrates primate visual mechanisms. As one of the latest versions in the YOLO series, YOLOv13 introduces further optimizations in its network architecture and feature fusion mechanisms compared with earlier models such as YOLOv8 and YOLOv10. These improvements not only significantly enhance its detection performance for small-scale targets and complex scenarios but also ensure its leading position in terms of the inference latency and model size. Experimental results on the MS COCO dataset demonstrate that YOLOv13-n achieves an mAP improvement of 3.0 and 1.5 percentage points over YOLOv11-n and YOLOv12-n, respectively.

To enhance the network’s capability for feature extraction of fresh tea leaves in complex tea garden environments, and to mimic the dynamic modulation process of the primate visual system regarding effective receptive field and channel gain, DynamicConv was employed to optimize the DS-C3k2 module and several convolutional layers within the network [15]. To improve multiscale feature extraction and discrimination, Efficient Mixed-pooling Channel Attention is introduced to emulate the primate visual system strategy of parallel global gain control and winner selection integration [16]. To improve the localization accuracy and robustness of the final model for small tea targets, a Scale-based Dynamic Loss is introduced to optimize the loss function [17]. This mechanism emulates the central concave mechanisms of the primate visual system by dynamically increasing the weighting of positional constraints for small-scale targets. This study aims to provide an efficient and stable yield detection solution to enhance the intelligent management and sustainable development of tea gardens. It also offers a feasible pathway for the deep integration of bionic vision and agricultural remote sensing.

## 2. Materials and Methods

### 2.1. Dataset Construction

To ensure both scientific rigor and practical applicability of the experimental data, this study selects Lincang City and Xishuangbanna Prefecture in Yunnan Province as the primary data collection areas. As key production regions for Yunnan large-leaf tea trees, both areas boast extensive cultivation histories and large-scale plantations. Their highly representative ecological and geographic characteristics ensure a robust and diverse set of samples for this investigation [18]. Moreover, Lincang and Xishuangbanna differ markedly in geographical environment, climatic conditions, and ecosystem structure, which offers a natural advantage for building a detection model with strong generalization ability and robustness.

To further enhance model generalization across diverse environments, a data acquisition strategy incorporating multiple time periods and viewing angles was employed, covering varied illumination, background complexity, and tea canopy density, thereby maximizing data representativeness and coverage [19]. Given that the height of the target plants is typically concentrated between 0.5 and 1.5 m, and that target recognition accuracy in images collected by unmanned aerial vehicles declines markedly when the shooting altitude exceeds 3 m, we restrict the flight altitude to the interval from 1.5 to 3 m to balance image resolution, target discernibility, and flight safety.

To ensure image resolution and consistency in data acquisition, this study adopts the DJI Mavic 2 Pro unmanned aerial vehicle as the remote sensing platform (DJI Innovations Co., Ltd., Shenzhen, China). Equipped with a 1-inch, 20-megapixel image sensor, the platform captures high-resolution imagery, which is crucial for precisely delineating tea canopy structures and leaf details. The device supports 4K HDR video recording and an adjustable aperture and it can automatically optimize exposure parameters in response to ambient illumination changes, ensuring stable image quality across different times of day and weather conditions. In addition, the Mavic 2 Pro integrates a 10-bit Dlog-M color mode and the OcuSync 2.0 high definition transmission system, which enhance color fidelity and dynamic range performance and strengthen the stability and anti-interference capability of long range data transmission. These features provide robust technical support for large-scale and long-term remote sensing monitoring across complex terrain, multiple time periods, and diverse climate conditions in this study.

In this study, a total of 2364 images of Yunnan large-leaf tea trees were collected, covering tender buds, one bud and one leaf, and one bud and two leaves. In the data preprocessing stage, 2147 high-quality images were selected to construct the original dataset after eliminating the blurred, repeated and invalid images. As shown in Figure 1, Figure 1A is the histogram of the number distribution of three types of fresh leaf labels. Figure 1B shows the size distribution of the label in the width and height dimensions, and the center points of all labels are normalized to the image center. Figure 1C shows the spatial distribution characteristics of the label in the image coordinate system. Figure 1D is the visualization of the width-to-height ratio distribution of the annotation box. Figure 1E further shows the detailed features of the label in the image [20].

The detailed data distribution of this study is presented in Table 1. A total of 7403 labels were annotated across 2147 original images. The collected original images were divided into training and test sets at a ratio of 8:2, with 80% used for five-fold cross validation training and 20% reserved for final model testing. In addition, an external validation dataset was independently collected from Yunnan Agricultural University in Kunming, Yunnan Province, consisting of 500 images with a total of 1864 labels.

### 2.2. Data Augmentation

To enhance the robustness and generalization capability of the yield detection model for Yunnan large-leaf tea trees in complex environments, this study further introduces HSV (Hue, Saturation, Value) color space perturbation, filtering-based data augmentation, brightness transformation, and random scaling along the X and Y axes, as illustrated in Figure 2. Among them, HSV augmentation applies random perturbations to the hue, saturation, and value channels to simulate variable illumination, color temperature shifts, and device-specific imaging differences. This enhances model stability in complex color environments. Mean Blur replaces pixel values with the mean of their neighbors, reducing interference from local details and simulating images at different resolutions or compression levels. This improves model stability under varying image quality conditions. Gaussian Blur smoothens the image through Gaussian kernel function, which can effectively simulate the slight blur caused by UAV in high-speed flight or imaging jitter, so as to improve the feature extraction ability of the model in low-definition images. Median Blur is primarily used to remove salt-and-pepper noise while preserving edge details and strengthening structural features of the image, thereby improving detection of edges and structural information in complex backgrounds. Brightness Adjustment randomly adjusts the overall brightness of the image to simulate the natural lighting conditions of different time periods such as early morning, noon and evening, so as to enhance the stability and generalization ability of the model in various lighting environments. Random Scaling on the X and Y axes performs the scale transformations of different ratios along the horizontal and vertical directions, expanding the scale diversity in the training samples and increasing model robustness and detection accuracy for targets under changes in viewpoint and size [21].

### 2.3. Improvements to the YOLOv13 Network

As one of the leading-edge networks in the field of object detection in recent years, YOLOv13 incorporates HyperACE (Hypergraph-based Adaptive Correlation Enhancement) while maintaining end-to-end detection efficiency, which substantially improves its capacity for modeling global multi-scale features [22]. However, when applied to the yield detection of Yunnan large-leaf tea trees, a task characterized by complex back-grounds, small target sizes, significant morphological variations, and severe occlusion in remote sensing imagery, the model still exhibits notable deficiencies in multi-scale feature representation and bounding box localization accuracy.

To address these limitations, this study adopts YOLOv13 as the baseline network. While preserving the network’s high efficiency and lightweight architecture, simulating the dynamic adjustment process of the visual system on the effective receptive field and channel gain. In Backbone and Neck, DynamicConv is used to optimize DS-C3k2 and some Conv to improve the network ‘s ability to characterize complex textures and diverse targets. Referring to the observation strategy of ‘global gain control and strong selection integration parallel’ of primate visual system, Efficient Mixed-pooling Channel Attention is introduced to optimize the head layer of the network to enhance the multi-scale feature extraction and discrimination ability of the network. Drawing on the foveation mechanism of primate vision, which performs fine perception for small targets and coarse representation for large targets, the loss function is optimized using Scale-based Dynamic Loss. This loss dynamically strengthens positional constraints for small targets while appropriately relaxing scale constraints, thereby enhancing localization accuracy and robustness. Simultaneously, the Scale-based Dynamic Loss is adaptively degraded to CIoU (Complete Intersection over Union) on large targets to maintain the stability of the overall detection. The detailed parameters of the improved DE-YOLOv13-S network are shown in Table 2, and the network structure is shown in Figure 3.

#### 2.3.1. DynamicConv Optimization

In the yield detection of Yunnan large-leaf tea trees, remote sensing images typically exhibit uneven target density, significant scale variation, and complex leaf textures [23]. However, the YOLOv13 network is highly dependent on static convolution in structure, so it is difficult to effectively expand the receptive field and feature expression ability without significantly increasing the computational overhead. In the face of dense tea leaves and fine-grained texture structures, its representation ability is limited.

In primate vision, the receptive fields of the center and periphery modulate responses in an antagonistic manner [24]. Together with contrast gain control, this mechanism enables the visual system to dynamically adjust the effective receptive fields and channel responses according to stimulus intensity and background complexity. Inspired by this mechanism, the present study employs DynamicConv to optimize the DS-C3k2 modules and part of the convolutional layers in the Backbone and Neck, in order to simulate the dynamic regulation of effective receptive fields and channel gain in the visual system, thereby enhancing the model’s feature representation capability and detection robustness in complex tea garden scenarios.

DynamicConv consists of three components, namely a coefficient generator, an expert kernel library, and a dynamic weight fusion and convolution process. The coefficient generator is defined by (1) and (2), where $X\in R^{C_{in}\times H\times W}$ and $\alpha \in R^{M}$. X denotes the input feature map, v denotes the vector obtained by global average pooling, α denotes the M dimensional normalized coefficient vector, $C_{in}$ is the number of input channels, H and W are the spatial dimensions of the input.(1)$$\alpha =softmax(MLP\left(v\right))$$(2)$$v=Pool\left(X\right)$$

The Expert Bank is shown in Equation (3), where $W_{i}$ is the standard convolution kernel weight tensor, $C_{out}$ is the number of output channels, K is the kernel size, and the total kernel size is K×K. The expert core library is the parameterized main body of DynamicConv, which is used to achieve the purpose of approximately linear expansion of capacity by M times and keeping FLOPs basically unchanged.(3)$$W_{i}\in R^{{C_{out}\times C}_{in}\times K\times K}$$

The dynamic weight fusion and convolution process is given in Equations (4) and (5), where $Y\in R^{C_{out}\times H^{’}\times W^{’}}$ and the symbol * denotes the discrete convolution operator. W’ denotes the equivalent convolution kernel obtained by coefficient based fusion of multiple expert kernels, Y denotes the output feature map, and $H^{’}$ and $W^{’}$ denote the spatial size of the output. In DynamicConv, this process contains two parts. The dynamic weight fusion linearly combines the coefficients α produced by the coefficient generator with the kernels in the expert bank to form an input adaptive equivalent kernel. The convolution process applies standard convolution to the fused kernel to produce feature mappings while preserving the standard convolution computation paradigm.(4)$$W’=\sum_{i=1}^{M}\alpha_{i}W_{i}$$(5)Y=X∗W’

As illustrated in Figure 4, the improved DynamicConv-C3k2 adopts an overall backbone branch architecture. For the input feature map, a 1×1 Conv first adjusts the channels to the target dimension. The adjusted features are then separated into a main path and a residual branch. The main path is formed by a series of DynamicConv-C3k blocks that extract deep semantic features, while the residual branch preserves the unaltered input as residual information to mitigate feature degradation. After the two parts are fused through feature concatenation, a 1×1 Conv performs channel consolidation and information integration to produce the final output features.

The architecture of DynamicConv-C3k is largely consistent with DynamicConv-C3k2. The first 1×1 Conv is used to compress the input channels to reduce the computational cost of subsequent modules. After feature separation, the main path is composed of several Bottleneck DynamicConv modules that strengthen the network ability to represent local structures and contextual semantics. Meanwhile, the residual branch uses a 1×1 convolution layer to preserve and adjust the original feature channels so that they align in dimension with the output of the main path, enabling efficient information compensation and fusion. The concatenated feature map then passes through a final 1×1 Conv layer to complete linear integration and to restore the channel count to the preset output dimension, producing the final output features.

#### 2.3.2. Efficient Mixed Pooling Channel Attention Optimization

In biological vision, information integration is achieved through two complementary processes: global baseline estimation of brightness and texture for gain control, and selective enhancement of peak responses [25]. Inspired by this mechanism, introduce a hybrid pooling strategy into YOLOv13 to extract both global and peak feature statistics. A lightweight interaction module along the channel dimension then generates attention weights, which are applied to the original features for channel recalibration. This process not only suppresses channel response overestimation caused by background noise but also highlights key responses related to object edges and textures, thereby improving the model’s discriminative capacity and robustness in complex tea garden scenarios.

At the algorithmic level, enhance the Efficient Channel Attention module pro-posed by Wang et al. by incorporating a joint strategy of Global Max Pooling and Global Average Pooling. The Efficient Mixed-pooling Channel Attention constructed after optimization is shown in Figure 5 [26]. In this module, Global Average Pooling provides a stable global estimate of image-wide brightness and texture features, whereas Global Max Pooling highlights the most discriminative peak responses, such as highlights at tea bud tips and edge reflections of tender leaves. After the fusion of the two, it can not only retain the strong response cues directly related to the characteristics of fresh leaves, but also effectively suppress the noise interference caused by local reflection, branch shadow and fine texture fluctuation, thus generating more robust channel attention weights and significantly improving the detection and counting reliability of small-scale tea buds in dense occlusion scenes.

After performing Global Max Pooling and Global Average Pooling on the input features and fusing them, a channel response vector *y* of length *C* is obtained. Cross-channel interaction is then established along the channel dimension. Here, k represents the neighborhood width, which is the size of the 1D convolutional kernel, as shown in Equation (6). In this, ψ represents the reverse mapping from the number of channels to the kernel size, γ is the mapping hyperparameter, and odd denotes an operator that selects the nearest odd number. b denotes the bias term of the mapping function.(6)$$k=\psi \left(C\right)={\left|\frac{l{og}_{2}C}{\gamma }+\frac{b}{\gamma }\right|}_{odd}$$

As the number of channels increases, the neighborhood width k also expands to cover a broader range of neighboring areas. To maintain consistent symmetric padding length, k is always set to the nearest odd number. A 1D convolution with kernel size k is applied to the channel response vector y, and the result is normalized using the Sigmoid function. EMCA (Efficient Mixed-pooling Channel Attention) then writes the resulting weights back to the original features to perform channel-wise weighting. This dynamically adjusts the amplitude of different channel responses while keeping the spatial dimensions and the number of channels unchanged.

#### 2.3.3. Scale-Based Dynamic Loss Optimization

In yield detection for Yunnan large-leaf tea trees, fresh tea leaves exhibit complex spatial distributions, featuring multi-scale variations, dense arrangements, deformations, and occlusions. They are also susceptible to interference from varying illumination and overlapping branch textures within the plantation. Under these conditions, the standard YOLOv13 model with the CIoU loss function couples the overlap, center distance, and aspect ratio using fixed weighting coefficients, thereby neglecting the varying sensitivities of targets to these constraints across different scales [27]. This static coupling approach can lead to gradient instability and regression bias, particularly for small or low-contrast targets like tea buds. Consequently, gradient oscillations and localization inaccuracies arise, resulting in fresh leaf counting errors that ultimately compromise yield detection accuracy.

In biological vision, the fovea is responsible for high precision fixation, where small objects or task relevant details receive greater attention and resources, while salient and easily distinguishable large objects are encoded more coarsely [28]. Correspondingly, in the loss design for object detection, emulate this principle by enabling the weights for the localization and scale terms to adapt dynamically based on the target size. For small objects, the localization constraints are strengthened and the scale term is moderately relaxed, yielding more stable gradients and lower regression bias. For sufficiently salient large objects, the weights return to a more balanced configuration to ensure overall training stability. Thus, the biological principle of foveal fine-coding and peripheral coarse representation is translated into a scale-aware dynamic reweighting strategy for the loss function. This approach enhances the localization accuracy and convergence quality for small targets while preserving balanced global performance.

At the algorithmic level, this study optimizes the loss function of YOLOv13 by introducing Scale-based Dynamic Loss without altering the original network architecture. For small-scale targets, the constraint on the localization term is dynamically increased and the constraint on the scale term is relaxed to enhance localization accuracy and robustness for very small objects. For large-scale targets, the formulation adaptively converges to the CIoU loss, thereby improving overall detection accuracy in complex tea plantation settings. The optimized Scale-based Dynamic Loss is given in Equations (7) and (9), where $L_{SDB}$ denotes the total loss, $L_{BS}$ and $L_{BL}$ correspond to the scale term loss and the localization term loss, and β is the dynamic weighting coefficient. $L_{BS}$ measures the consistency of overlap and aspect ratio between the predicted and ground truth boxes, and $L_{BL}$ characterizes the alignment of their centers through a normalized center distance. Together they provide complementary constraints on target shape and position. IoU denotes the intersection over union between the predicted and ground truth boxes. v is the measure of aspect ratio consistency, and α is the weight adjusted by IoU and v. $b_{p}$ and $b_{gt}$ denote the centers of the predicted and ground truth boxes, ρ denotes the Euclidean distance, and c denotes the diagonal length of the minimum enclosing rectangle that covers both boxes.(7)$$L_{SDB}=\beta_{L_{BS}}L_{BS}+\beta_{L_{BL}}L_{BL}$$(8)$$L_{BS}=1-IoU+\alpha v$$(9)$$L_{BL}=\frac{\rho^{2}(b_{p},b_{gt})}{c^{2}}$$

The computation of the dynamic weighting coefficients is given in Equations (10) and (11), where $R_{OC}$ denotes the scale factor that maps the target area at the feature map scale back to the original image scale. $w_{o}$ and $h_{o}$ are the width and height of the original image, $w_{c}$ and $h_{c}$ are the width and height of the current feature map. δ denotes the adjustable upper bound of the dynamic amplitude, with δ∈(0,1]. $B_{gt}$ is the area of the ground truth box, and $B_{gt}^{max}$ denotes the upper bound for Infrared Small Target size under the Society of Photo Optical Instrumentation Engineers standard, which is 81 pixels. When the area of $B_{gt}$ exceeds 81, the values of $\beta_{L_{BS}}$ and $\beta_{L_{BL}}$ automatically become 1, and the loss function automatically degenerates to the CIoU loss [17].(10)$$R_{OC}=\frac{w_{o}\times h_{o}}{w_{c}\times h_{c}}$$(11)$$\beta_{B}=min(\frac{B_{gt}}{B_{gt}^{max}}\times R_{OC}\times \delta ,\delta )$$

### 2.4. Five-Fold Cross Validation

To enhance the generalization ability of YOLOv13 in the yield detection task for Yunnan large leaf tea trees and to reduce performance fluctuations due to data partitioning, this study further added five-fold cross-validation to optimize the network training stage [29]. As shown in Figure 6, compared to traditional training and validation splits, five-fold cross-validation randomly divides the dataset into five subsets of approximately equal size. In each round of experimentation, one subset is selected as the validation set, and the remaining four subsets are used for training. This process is repeated five times, with training and validation performed in a cyclic manner. This method effectively reduces result bias caused by sample distribution differences, maximizes the utilization of limited sample resources, and enhances the comprehensiveness of feature learning and the robustness of result evaluation.

### 2.5. Evaluation Metrics

To comprehensively evaluate the improved YOLOv13 on the yield detection task for Yunnan large leaf tea trees, we assess accuracy, detection capability, and overall performance, using Precision, Recall, F1-Score, and mAP as metrics, as given in Equations (12)–(15) [30]. Precision denotes the proportion of detected tea bud targets that are truly tea buds. Recall denotes the proportion of all true tea bud targets that are correctly detected. F1-Score is the harmonic mean of precision and recall and provides a balanced summary of both. $T_{P\;}$(True Positives) is the number of tea buds correctly detected by the model, $F_{P}$ (Positive Falses) is the number of false detections, and $F_{N}$ (False Negatives) is the number of missed detections. *AP* (Average Precision) represents the area under the Precision-Recall curve of a single category under a given IoU threshold, which is used to measure the comprehensive detection performance of the model under different confidence thresholds [31]. *mAP* is the arithmetic mean of all categories of APs, which is used to evaluate the overall performance of the model in multi-category target detection tasks. This study sets the confidence threshold during the inference phase uniformly to 0.25 and uses mAP@0.5 as the evaluation metric for detection performance [32,33,34,35,36].(12)$$Precision=\frac{T_{P}}{T_{P}+F_{P}}$$(13)$$Recall=\frac{T_{P}}{T_{P}+F_{N}}$$(14)$$F1=2*\frac{Precision*Recall}{Precision+Recall}$$(15)$$mAP=\frac{\sum_{i=1}^{k}{AP}_{i}}{k}$$

## 3. Results and Analysis

To ensure the rigor of the results, all experiments were conducted under a unified hardware and software environment. The hardware configuration is Windows 11 operating system, NVIDIA GeForce RTX 4060 Ti (16 GB video memory) graphics card, 16 GB DDR4 3200 MHz memory, Kingston NV2 1 TB SSD, NVIDIA 561.09 driver, CUDA 12.6 version. The software environment is Python 3.12 and PyCharm 2023, and the Batch size and Epoch are set to 16 and 500, respectively, during the training process. The initial learning rate is uniformly set to 0.01, with the optimizer configured as SGD and a weight decay of 0.001.

### 3.1. Analysis of Model Results

The loss function is one of the core components in the deep learning model, which is mainly used to quantitatively evaluate the deviation between the predicted value of the model and the real label [37,38,39,40]. In the yield detection task of large-leaf tea trees in Yunnan, the loss value can effectively reflect the accuracy of the model in target location and category recognition. The lower the loss value, the more accurate the model’s detection of tea tree yield targets, and the better the overall performance. As shown in Figure 7, the loss of DE-YOLOv13-S converges slightly faster than that of the original YOLOv13. On the training set, the Box Loss, Cls Loss, and DFL Loss of DE-YOLOv13-S stabilize below 0.65, 0.52, and 1.15, which are decreases of 18.75%, 3.70%, and 2.54% compared with the original YOLOv13 values of 0.80, 0.54, and 1.18. On the test set, Box Loss, Cls Loss, and DFL Loss stabilize below 0.75, 0.60, and 1.24, which are decreases of 18.48%, 14.29%, and 7.46% compared with the original YOLOv13 values of 0.92, 0.70, and 1.34. These results show that the optimized DE-YOLOv13-S improves detection accuracy and generalization.

As shown in Figure 8, the Precision, Recall, and F1 of the DE-YOLOv13-S network are 82.32%, 87.26%, and 84.72%, which are improvements of 3.78%, 2.04%, and 2.97% over the original YOLOv13. These results indicate that the improved model has stronger target recognition capability for fresh tea leaf detection and can localize and identify target regions more accurately.

The confusion matrix can visually present the classification performance of the model in the research on the yield detection of Yunnan big leaf tea tree [41]. As shown in Figure 9, the rows correspond to the real tea categories, and the columns correspond to the prediction results of the model. The deeper the color of the diagonal elements and the closer the value to 1, the higher the recognition accuracy of the model for the fresh leaves of this category. Non-diagonal elements reflect the degree of confusion between different categories. The non-zero elements at the lower left of the diagonal correspond to the undetected fresh tea leaves, and the non-zero elements at the upper right correspond to the wrongly detected fresh tea leaves. The results show clear improvements over the original YOLOv13. The detection accuracy for Tender bud increases by 1 percentage point, for One bud with one leaf by 3 percentage points, and for One bud with two leaves by 4 percentage points. DE YOLOv13 S effectively reduces confusion between small-scale targets and visually similar categories, thereby improving the reliability of yield detection.

### 3.2. Ablation Experiments

To systematically evaluate the independent improvement effect of structural improvement inspired by each sensing mechanism on model performance, this study further designed and carried out ablation experiments. By introducing DynamicConv, Efficient Mixed-pooling Channel Attention and Scale-based Dynamic Loss one by one, the influence of each module on the target detection accuracy, recall rate, mAP, FLOPs, Parameters and Gradients indicators is compared and analyzed to clarify the performance gain of different perception strategies on the detection task of Yunnan large-leaf tea leaves. The results of ablation experiments are shown in Table 3. When the DynamicConv optimization is introduced separately, although the Parameters and Gradients are increased by 2.06 M, the FLOPs are reduced by 0.2 G, and the Precision, Recall, and mAP are increased by 1.7%, 1.1%, and 1.36%, respectively. Experimental tests show that DynamicConv can expand the effective receptive field and adjust channel gain, thereby better representing fine-grained textures while reducing redundant static computations. With only Efficient Mixed-pooling Channel Attention, changes in parameters, gradients, and FLOPs remain within 1 percent, while precision, recall, and mAP increase by 2.74 percent, 0.09 percent, and 0.63 percent. The global gain estimation from GAP and GMP, combined with adaptive one-dimensional channel interaction, can to some extent suppress pseudo-responses such as glare and branch shadows, while selectively amplifying discriminative channels, thereby effectively improving precision. Scale-based Dynamic Loss keeps parameters, gradients, and FLOPs unchanged, yet raises precision, recall, and mAP by 0.92 percent, 1.72 percent, and 0.85 percent. The improved loss function can dynamically reweight the localization and scale terms according to object size, thereby alleviating regression bias for small objects and stabilizing gradient propagation. With the multi strategy configuration, parameters and gradients of DE-YOLOv13-S increase by 2.06 M while computational complexity drops by 0.2 G FLOPs, and precision, recall, and mAP rise by 3.78 percent, 2.04 percent, and 3.35 percent. The overall detection performance is significantly improved compared with the original network, which can be attributed to the complementary effects of the three proposed enhancements. DynamicConv effectively expands the effective receptive field and improves multi-scale representation, while Efficient Mixed-Channel Attention generates more robust channel attention weights. In addition, the dynamic adjustment of localization constraints for small objects in Scale-based Dynamic Loss Optimization further enhances the final model’s localization accuracy for small targets. In terms of detection speed, DynamicConv optimization benefits from the reduction in FLOPs and improves the FPS by 2.40 compared with the original network. The scale adaptive constraints introduced by Scale-based Dynamic Loss enhance the regression stability of small objects, indirectly reducing computational overhead and increasing the FPS by 1.58 over the original network. In contrast, Efficient Mixed pooling Channel Attention introduces a small amount of channel interaction, resulting in a slight decrease of 0.38 FPS. After integrating all improvements, the overall FPS of the model shows a net increase of 1.99, indicating a notable improvement in detection speed.

### 3.3. Model Comparison Experiments

To comprehensively evaluate the overall performance of the DE-YOLOv13-S network in the task of tea tree yield detection, this study designed five groups of comparative experiments, using representative object detection models such as YOLOv13, SSD, CornerNet, and RT-DETR as baselines, with model testing conducted on the final test set. The test results are shown in Table 4. Compared with YOLOv13, SSD, CornerNet and RT-DETR, the Precision of DE-YOLOv13-S increased by 3.78%, 8.13%, 12.03% and 4.14%, respectively. Recall increased by 2.04%, 9.77%, 9.9% and 3.6%, respectively, F1 increased by 2.98%, 8.92%, 11.06% and 3.89%, respectively, and mAP increased by 3.35%, 13.08%, 15.41% and 5.17%, respectively. For tender bud recognition, DE-YOLOv13-S achieves AP improvements of 2.59%, 13.56%, 16.44%, and 5.35%. For one bud and one leaf recognition, the AP values are improved by 3.08%, 14.79%, 16.29%, and 5.39%. For one bud and two leaves recognition, the AP values are improved by 4.38%, 10.89%, 13.50%, and 4.78%. The results show that DE-YOLOv13-S has better performance than mainstream detection models such as YOLOv13, SSD, CornerNet and RT-DETR.

To further verify the detection ability of DE-YOLOv13-S in the tea yield detection task and verify its overall advantages in the tea yield detection task, this study conducted external verification for low-light conditions, image, complex backgrounds, and object occlusion [42]. The external verification data were collected from Yunnan Agricultural University in Kunming, Yunnan Province. The results are shown in Figure 10. DE-YOLOv13-S still maintains high detection accuracy in a variety of complex environments, and can accurately capture and locate the target of fresh tea leaves. Compared with the original YOLOv13, the undetected rate and confidence are significantly reduced.

## 4. Conclusions and Discussion

To overcome challenges such as variable target scales, complex backgrounds, image blur, and severe occlusion in yield detection for Yunnan large-leaf tea trees, this study introduces DE-YOLOv13-S. This novel deep learning network is inspired by primate visual mechanisms and specifically designed for this application. Built upon the YOLOv13 architecture, the model incorporates DynamicConv to optimize the DS-C3k2 module and select convolutional layers, emulating the dynamic modulation of effective receptive fields and channel gain observed in biological vision. Efficient Mixed pooling Channel Attention is introduced to emulate the observation strategy of primate vision. Scale-based Dynamic Loss is used to optimize the loss function and simulate the foveation mechanism of primates. The results show that:

(1) The optimized DE-YOLOv13-S model demonstrates improved detection accuracy and generalization, as evidenced by its loss dynamics. On the training set, Box Loss, Cls Loss, and DFL Loss stabilize below 0.65, 0.52, and 1.15, representing decreases of 18.75 percent, 3.70 percent, and 2.54 percent from the original YOLOv13 values of 0.80, 0.54, and 1.18. On the test set, Box Loss, Cls Loss, and DFL Loss stabilize below 0.75, 0.60, and 1.24, which are reductions of 18.48 percent, 14.29 percent, and 7.46 percent from the original values of 0.92, 0.70, and 1.34.

(2) The ablation results show that parameters and gradients of DE YOLOv13 S increase by 2.06 M, while computational complexity decreases by 0.2 G FLOPs. At the same time, precision, recall, and mAP rise by 3.78 percent, 2.04 percent, and 3.35 percent, yielding overall detection performance clearly superior to the original model.

(3) From the model comparison, the experiment indicates that DE YOLOv13 S outperforms mainstream detectors including YOLOv13, SSD, CornerNet, and RT DETR. Precision increases by 3.78 percent, 8.13 percent, 12.03 percent, and 4.14 percent. Recall increases by 2.04 percent, 9.77 percent, 9.9 percent, and 3.6 percent. F1 increases by 2.98 percent, 8.92 percent, 11.06 percent, and 3.89 percent. mAP increases by 3.35 percent, 13.08 percent, 15.41 percent, and 5.17 percent. External validation under challenging conditions confirms these findings, with DE-YOLOv13-S accurately localizing fresh tea leaves, significantly reducing missed detections, and yielding higher detection confidence compared to the original YOLOv13.

In summary, the DE-YOLOv13-S network offers an efficient and robust solution for yield detection, contributing to more intelligent management and sustainable development in tea cultivation. It also presents a viable pathway for the deep integration of bionic vision and agricultural remote sensing. It should be noted that although this study includes an external validation set, the data collection sites remain relatively limited, and the generalizability to other regions, tea varieties, and extreme climatic conditions requires further validation. Moreover, while the improved network demonstrates notable improvements under dense occlusion, strong reflections, and extreme illumination, its robustness still shows a certain degree of decline. In future work, the team will further explore multimodal sensor data fusion, cross-region model transfer capability, and few shot learning strategies, while continuously expanding the study areas and target scope to broaden the applicability and adaptability of this method to the intelligent detection of diverse economic crops. In addition, we plan to further introduce strategies such as image augmentation to perform additional processing on the input images, so as to enhance the practicality of the model under extreme illumination and other challenging environmental conditions.

## Figures and Tables

**Figure 1 biomimetics-10-00724-f001:**
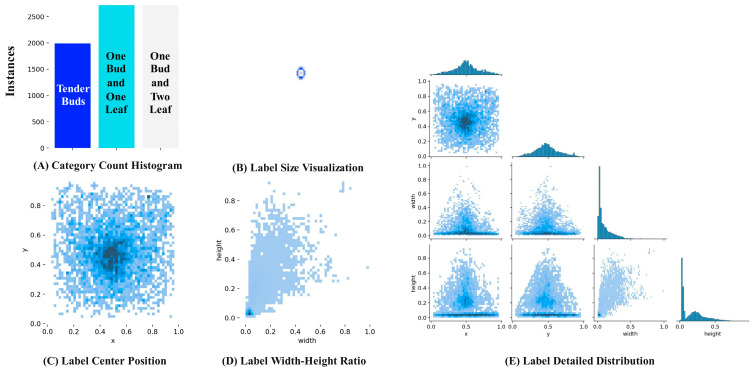
Dataset label visualization.

**Figure 2 biomimetics-10-00724-f002:**
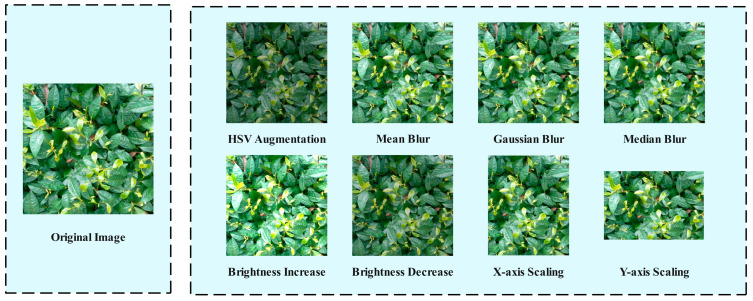
Data augmentation.

**Figure 3 biomimetics-10-00724-f003:**
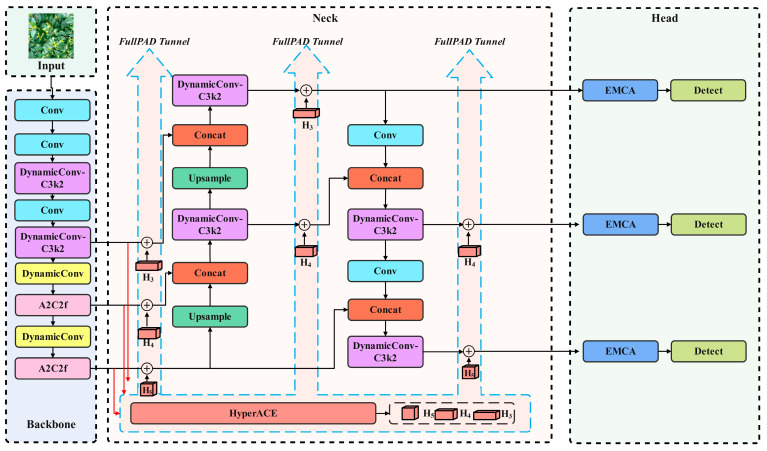
Improved YOLOv13 network.

**Figure 4 biomimetics-10-00724-f004:**
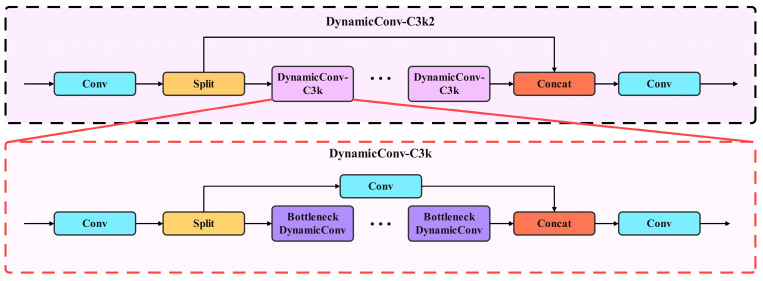
DynamicConv-C3k2 and DynamicConv-C3k.

**Figure 5 biomimetics-10-00724-f005:**
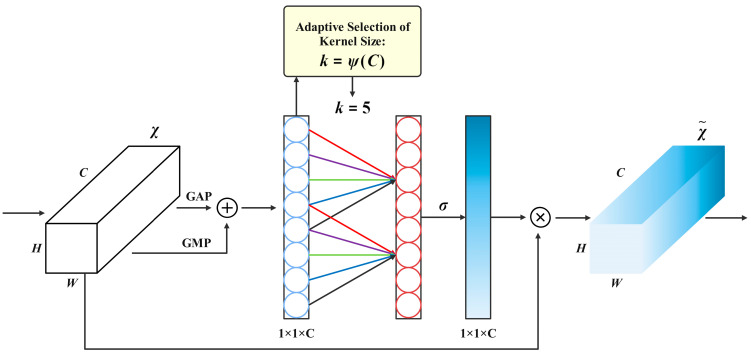
Efficient Mixed pooling Channel Attention.

**Figure 6 biomimetics-10-00724-f006:**
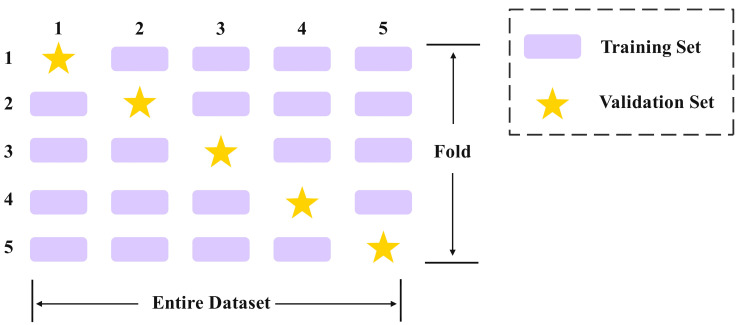
Five-fold Cross Validation.

**Figure 7 biomimetics-10-00724-f007:**
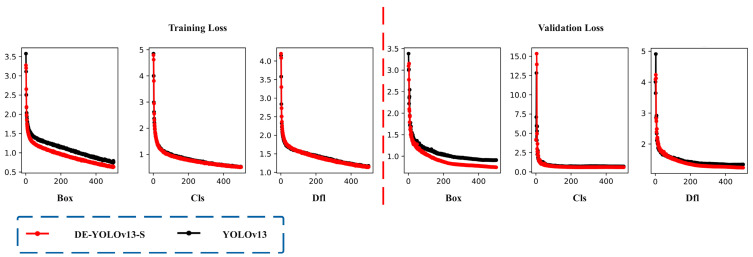
Loss function comparison chart. Note: The X axis represents the number of training epochs, and the Y axis represents the loss value.

**Figure 8 biomimetics-10-00724-f008:**
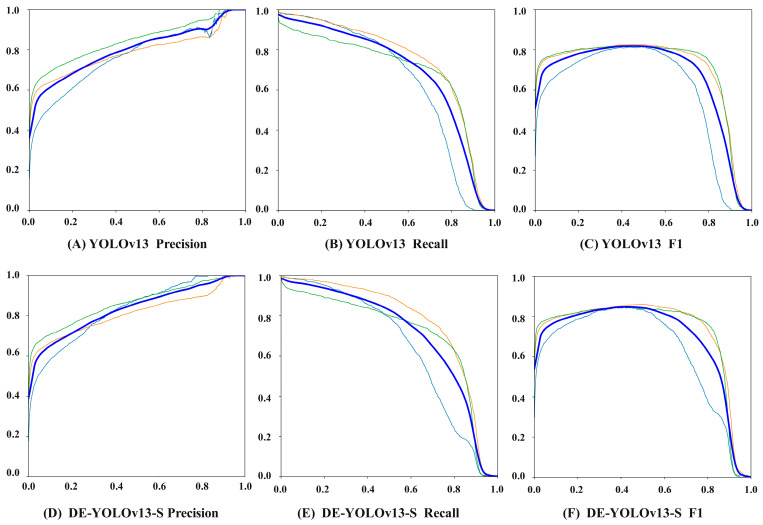
Precision Recall and F1 comparison chart. Note: In all subfigures, the X axis represents the confidence threshold. The Y axis of (**A**,**D**) corresponds to Precision, that of (**B**,**E**) corresponds to Recall, and that of (**C**,**F**) corresponds to F1. The thin blue line represents one bud with two leaves, the thin red line represents tender buds, the thin green line represents one bud with one leaf, and the thick blue line represents all categories.

**Figure 9 biomimetics-10-00724-f009:**
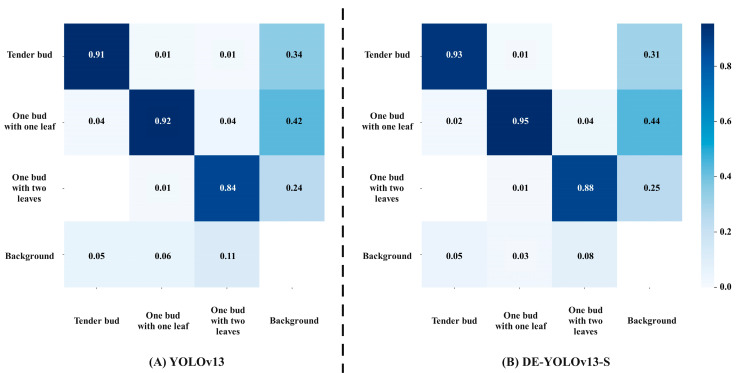
Confusion matrix comparison chart.

**Figure 10 biomimetics-10-00724-f010:**
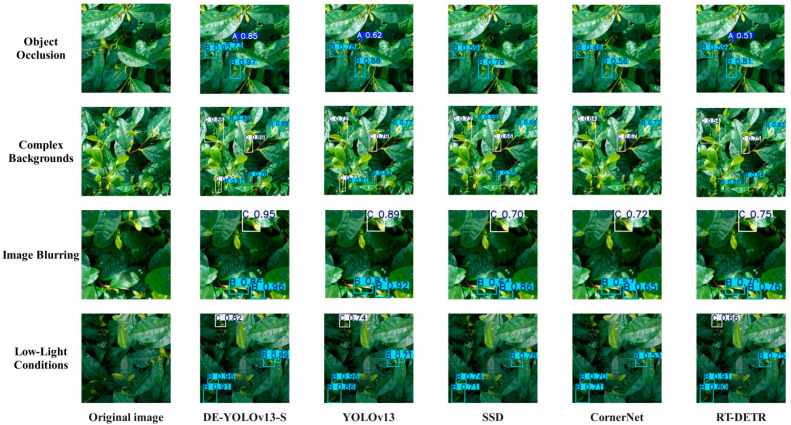
External validation comparison chart.

**Table 1 biomimetics-10-00724-t001:** Dataset label distribution.

Tea Category	Original Image			External Validation Set
	Total Number of Labels	Training Set (Five Fold Cross Validation)	Test Set	
Tender Buds	1986	1612	374	545
One Bud and One Leaf	2723	2286	437	645
One Bud and Two Leaf	2694	2246	448	674

**Table 2 biomimetics-10-00724-t002:** Detailed parameters of the DE-YOLOv13-S network.

ID	From	Params	Module	Arguments
0	−1	464	Conv	[3, 16, 3, 2]
1	−1	2368	Conv	[16, 32, 3, 2, 1, 2]
2	−1	15,940	DynamicConv-C3k2	[32, 64, 1, False, 0.25]
3	−1	9344	Conv	[64, 64, 3, 2, 1, 4]
4	−1	63,108	DynamicConv-C3k2	[64, 128, 1, False, 0.25]
5	−1	590,596	DynamicConv	[128, 128, 3, 2]
6	−1	174,720	A2C2f	[128, 128, 2, True, 4]
7	−1	1,180,676	DynamicConv	[128, 256, 3, 2]
8	−1	677,120	A2C2f	[256, 256, 2, True, 1]
9	[4, 6, 8]	273,536	HyperACE	[128, 128, 1, 4, True, True, 0.5, 1, ‘both’]
10	−1	0	Upsample	[None, 2, ‘nearest’]
11	9	33,280	DownsampleConv	[128]
12	[6, 9]	1	FullPAD_Tunnel	[]
13	[4, 10]	1	FullPAD_Tunnel	[]
14	[8, 11]	1	FullPAD_Tunnel	[]
15	−1	0	Upsample	[None, 2, ‘nearest’]
16	[−1, 12]	0	Concat	[1]
17	−1	175,368	DynamicConv-C3k2	[384, 128, 1, True]
18	[−1, 9]	1	FullPAD_Tunnel	[]
19	17	0	Upsample	[None, 2, ‘nearest’]
20	[−1, 13]	0	Concat	[1]
21	−1	48,264	DynamicConv-C3k2	[256, 64, 1, True]
22	10	8320	Conv	[128, 64, 1, 1]
23	[21, 22]	1	FullPAD_Tunnel	[]
24	−1	36,992	Conv	[64, 64, 3, 2]
25	[−1, 18]	0	Concat	[1]
26	−1	150,792	DynamicConv-C3k2	[192, 128, 1, True]
27	[−1, 9]	1	FullPAD_Tunnel	[]
28	26	147,712	Conv	[128, 128, 3, 2]
29	[−1, 14]	0	Concat	[1]
30	−1	600,584	DynamicConv-C3k2	[384, 256, 1, True]
31	[−1, 11]	1	FullPAD_Tunnel	[]
32	23	4	EMCA	[64]
33	27	4	EMCA	[128]
34	31	4	EMCA	[256]
35	[32, 33, 34]	431,257	Detect	[3, [64, 128, 256]]

**Table 3 biomimetics-10-00724-t003:** Ablation experiment results.

Model	Precision (%)	Recall (%)	mAP (%)	FLOPs (G)	Parameters	Gradients	FPS
YOLOv13	78.54	85.22	88.71	6.4	2,460,496	2,460,480	62.11
D-YOLOv13	80.25	86.32	90.07	6.2	4,620,448	4,620,432	64.51
E-YOLOv13	81.28	85.31	89.34	6.4	2,460,508	2,460,492	61.73
YOLOv13-S	79.46	86.94	89.56	6.4	2,460,496	2,460,480	63.69
DE-YOLOv13	82.12	85.58	90.97	6.2	4,620,460	4,620,444	63.29
D-YOLOv13-S	82.03	87.01	91.18	6.2	4,620,448	4,620,432	64.94
E-YOLOv13-S	81.31	87.12	90.92	6.4	2,460,508	2,460,492	62.50
DE-YOLOv13-S	82.32	87.26	92.06	6.2	4,620,460	4,620,444	64.10

Note: D denotes DynamicConv optimization, E denotes Efficient Mixed pooling Channel Attention optimization, and S denotes Scale-based Dynamic Loss optimization.

**Table 4 biomimetics-10-00724-t004:** Model comparison experiment results.

Model	Precision (%)	Recall (%)	F1 (%)	A_AP (%)	B_AP (%)	C_AP (%)	mAP (%)
DE-YOLOv13-S	82.32	87.26	84.72	93.11	94.62	88.45	92.06
YOLOv13	78.54	85.22	81.74	90.52	91.54	84.07	88.71
SSD	74.19	77.49	75.80	79.55	79.83	77.56	78.98
CornerNet	70.29	77.36	73.66	76.67	78.33	74.95	76.65
RT-DETR	78.18	83.66	80.83	87.76	89.23	83.67	86.89

Note: A represents Tender Buds. B represents One Bud and One Leaf. C represents One Bud and Two Leaves.

## Data Availability

The original code presented in the study are openly available in IEEE DataPort at https://dx.doi.org/10.21227/drd6-b843.

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
