# Peer review of "DE-YOLOv13-S: Research on a Biomimetic Vision-Based Model for Yield Detection of Yunnan Large-Leaf Tea Trees"

_biomimetics, 2025, doi:10.3390/biomimetics10110724_

Round 1
Reviewer 1 Report
Comments and Suggestions for Authors
The manuscript proposes an impactful technique however it needs some improvement before acceptance,
@ Several variables and hyperparameters (e.g., A, k in Equation 6) are not adequately defined or justified, making the mathematical formulation unclear.
@ The threshold of 81 pixels used in the dynamic loss adjustment lacks explanation or citation.
@ The evaluation metrics section does not mention confidence thresholds or the specific IoU thresholds used for mAP computation.
@ Important training parameters such as batch size, learning rate, optimizer, training epochs, and hardware used are missing.
@ Visual comparisons (e.g., attention maps, detection outputs) between baseline and improved model are not provided.
@ While biological strategies are mentioned, there is limited scientific explanation or evidence supporting their relevance to the model design.
@ To enhance the methodological depth and ensure alignment with current advancements in the field, the authors are advised to update and expand their reference list with more recent and technically relevant literature. Incorporating studies focused on improved YOLO architectures, lightweight detection frameworks, attention optimization, and dynamic loss strategies would provide stronger support for the proposed methods and better situate the work within the broader research landscape. The following DOIs are recommended for inclusion: doi.org/10.3390/jmse12101748, doi.org/10.1109/TNNLS.2024.3487833, doi.org/10.3390/rs16101775, doi.org/10.1016/j.measurement.2025.117852, doi.org/10.1007/s13369-025-10573-4, doi.org/10.32604/cmc.2024.052582, doi.org/10.1007/s11802-025-6029-2, doi.org/10.1016/j.asoc.2025.113260. Furthermore, the authors are encouraged to independently search for additional recent works from 2024 and 2025 that address similar technical challenges, to ensure the manuscript reflects the most up-to-date research and maintains scholarly completeness.
Author Response
Thanks very much for your time to review this manuscript. I really appreciate you’re your comments and suggestions. We have considered these comments carefully and tried our best to address every one of them. And the corresponding part in the text has been modified using red font.
- Several variables and hyperparameters (e.g., A, k in Equation 6) are not adequately defined or justified, making the mathematical formulation unclear.
Modification instructions: We sincerely appreciate your careful comments on the formulation of equations. We fully agree with your suggestion that the definitions of symbols should be more rigorous and complete. Accordingly, we have thoroughly reviewed and revised all equations throughout the manuscript to ensure that each letter and operator is clearly defined upon its first appearance.
“After performing Global Max Pooling and Global Average Pooling on the input features and fusing them, a channel response vector y of length C is obtained. Cross-channel interaction is then established along the channel dimension. Here, represents the neighborhood width, which is the size of the 1D convolutional kernel, as shown in Equation 6. In this, represents the reverse mapping from the number of channels to the kernel size, is the mapping hyperparameter, and denotes an operator that selects the nearest number. denotes the bias term of the mapping function.”
(6)
- The threshold of 81 pixels used in the dynamic loss adjustment lacks explanation or citation.
Modification instructions: We sincerely appreciate the reviewer’s reminder. The 81-pixel threshold in the manuscript is set as the upper bound according to the SPIE (Society of Photo-Optical Instrumentation Engineers) definition standard for Infrared Small Target size, and it follows the setting adopted in reference [17]. We have added the corresponding citation to [17] in that section. Thank you again for the helpful guidance.
- The evaluation metrics section does not mention confidence thresholds or the specific IoU thresholds used for mAP computation.
Modification instructions: We sincerely appreciate your valuable comment. We fully agree with your suggestion and have added the corresponding clarification in Section 2.5 Evaluation Metrics of the revised manuscript: “This study sets the confidence threshold during the inference phase uniformly to 0.25 and uses mAP@0.5 as the evaluation metric for detection performance.”
- Important training parameters such as batch size, learning rate, optimizer, training epochs, and hardware used are missing.
Modification instructions: We sincerely appreciate you for pointing out this critical issue. We fully acknowledge and accept your suggestion. In response, we have provided complete and standardized clarifications in the manuscript regarding the batch size, learning rate, optimizer, number of training epochs, and hardware specifications, to ensure that the experimental settings are transparent, reproducible, and easily verifiable by peers.“To ensure the rigor of the results, all experiments were conducted under a unified hardware and software environment. The hardware configuration is Windows 11 oper-ating system, NVIDIA GeForce RTX 4060 Ti (16 GB video memory) graphics card, 16 GB DDR4 3200 MHz memory, Kingston NV2 1TB SSD, NVIDIA 561.09 driver, CUDA 12.6 version. The software environment is Python 3.12 and PyCharm 2023, and the Batch size and Epoch are set to 16 and 500 respectively during the training process. The initial learning rate is uniformly set to 0.01, with the optimizer configured as SGD and a weight decay of 0.001.”
- Visual comparisons (e.g., attention maps, detection outputs) between baseline and improved model are not provided.
Modification instructions: We sincerely appreciate your valuable comment. To provide a more rigorous validation of our model, we have included visual comparisons of different models in the external validation section of 3.3 Model Comparison Experiments. It is possible that these images did not display properly in the review system due to their large file size. We have now adjusted the figures accordingly and reinserted them below in this response for your kind review.
- While biological strategies are mentioned, there is limited scientific explanation or evidence supporting their relevance to the model design.
Modification instructions: We sincerely thank you for your insightful comments on the correspondence between biological inspiration and algorithmic design. We fully agree with this opinion and have revised the manuscript by adding detailed explanations for each improved module, elaborating the connection from primate visual mechanisms to algorithmic implementation.
The supplementary content is as follows: “In primate vision, the receptive fields of the center and periphery modulate re-sponses in an antagonistic manner. Together with contrast gain control, this mechanism enables the visual system to dynamically adjust the effective receptive fields and channel responses according to stimulus intensity and background complexity. Inspired by this mechanism, the present study employs DynamicConv to optimize the DS-C3k2 modules and part of the convolutional layers in the Backbone and Neck, in order to simulate the dynamic regulation of effective receptive fields and channel gain in the visual system, thereby enhancing the model’s feature representation capability and detection robustness in complex tea garden scenarios.
In biological vision, information integration is achieved through two complemen-tary processes: global baseline estimation of brightness and texture for gain control, and selective enhancement of peak responses. Inspired by this mechanism, the present study introduces hybrid pooling into YOLOv13 to extract both global and peak statistics. A lightweight interaction is then constructed along the channel dimension to generate channel attention weights, which are reintegrated into the original features for channel recalibration. This process not only suppresses channel response overestimation caused by background noise but also highlights key responses related to object edges and textures, thereby improving the model’s discriminative capacity and robustness in complex tea garden scenarios. At the algorithmic level, this study builds upon the Efficient Channel Attention proposed by Qilong Wang et al. by introducing a joint strategy of Global Max Pooling and Global Average Pooling. The Efficient Mixed-pooling Channel Attention construct-ed after optimization is shown in Figure 5.
In biological vision, the fovea is responsible for high precision fixation, where small objects or task relevant details receive greater attention and resources, while salient and easily distinguishable large objects are encoded more coarsely. Correspondingly, in the loss design for object detection, this study adopts a similar principle by allowing the weights of the localization and scale terms to adapt dynamically to the actual object size. For small objects, the localization constraints are strengthened and the scale term is moderately relaxed, yielding more stable gradients and lower regression bias. For sufficiently salient large objects, the weights return to a more balanced configuration to ensure overall training stability. In this way, the biological allocation principle of finer encoding in the fovea and coarser representation in the periphery is translated into a scale aware dynamic reweighting strategy for the loss function, thereby improving the localization accuracy and convergence quality of small objects while maintaining balanced global performance. At the algorithmic level, this study optimizes the loss function of YOLOv13 by introducing Scale based Dynamic Loss without altering the original network architecture.”
- To enhance the methodological depth and ensure alignment with current advancements in the field, the authors are advised to update and expand their reference list with more recent and technically relevant literature. Incorporating studies focused on improved YOLO architectures, lightweight detection frameworks, attention optimization, and dynamic loss strategies would provide stronger support for the proposed methods and better situate the work within the broader research landscape. The following DOIs are recommended for inclusion: doi.org/10.3390/jmse12101748, doi.org/10.1109/TNNLS.2024.3487833, doi.org/10.3390/rs16101775, doi.org/10.1016/j.measurement.2025.117852, doi.org/10.1007/s13369-025-10573-4, doi.org/10.32604/cmc.2024.052582, doi.org/10.1007/s11802-025-6029-2, doi.org/10.1016/j.asoc.2025.113260. Furthermore, the authors are encouraged to independently search for additional recent works from 2024 and 2025 that address similar technical challenges, to ensure the manuscript reflects the most up-to-date research and maintains scholarly completeness.
Modification instructions: Thank you very much for your valuable suggestion. We have carefully reviewed and cited the recommended references in the revised manuscript, and updated and standardized other related citations. In addition, we have included several recent studies published in 2024 and 2025 that are closely related to the technical approach of this research, to ensure that the paper fully reflects the latest developments in the field and maintains academic integrity.

Reviewer 2 Report
Comments and Suggestions for Authors
1. The paper uses YOLOv13 as the baseline model, but this version is not well-established in the literature compared to YOLOv5–YOLOv10. The authors should provide a stronger justification for choosing YOLOv13, including references or a brief comparison with other YOLO versions to establish its relevance and superiority as a starting point.
2. While the model is described as “biomimetic,” the connection between primate visual mechanisms (e.g., foveation, gain control) and the technical improvements (DynamicConv, EMCA, Scale-based Loss) is somewhat superficial. The authors should more clearly articulate how each bio-inspired concept directly translates into the algorithmic design.
3. The ablation study results are presented in a dense table with many model variants, but the narrative does not sufficiently explain the contribution of each component. A clearer discussion of why certain combinations perform better—especially the interaction between modules—would strengthen the technical contribution.
4. The paper claims practical utility for tea garden management but does not discuss deployment constraints (e.g., computational cost on edge devices, inference speed). A discussion on real-time performance or scalability would enhance the practical relevance of the work.
5. The manuscript should be carefully reviewed to ensure all external sources are properly cited and that the text presents original synthesis and phrasing. While some similarity is expected in technical writing due to standard terminology and methods, the authors must verify that all borrowed ideas and text are appropriately referenced to maintain academic integrity. A thorough check for correct paraphrasing and quotation is recommended before publication.
The manuscript is generally well-written, with clear technical descriptions and coherent structure. However, there are occasional grammatical inconsistencies, awkward phrasings, and punctuation issues (e.g., missing articles, inconsistent tense usage). A thorough proofreading by a native English speaker is recommended to improve readability and polish.
Author Response
Thanks very much for your time to review this manuscript. I really appreciate you’re your comments and suggestions. We have considered these comments carefully and tried our best to address every one of them. And the corresponding part in the text has been modified using red font.
- The paper uses YOLOv13 as the baseline model, but this version is not well-established in the literature compared to YOLOv5–YOLOv10. The authors should provide a stronger justification for choosing YOLOv13, including references or a brief comparison with other YOLO versions to establish its relevance and superiority as a starting point.
Modification instructions: We sincerely appreciate your professional suggestion and fully concur with your opinion. Accordingly, we have revised the manuscript to incorporate additional content that reinforces the rationale for selecting YOLOv13 as the baseline model.
The supplementary content is as follows: “As one of the latest versions in the YOLO series, YOLOv13 introduces further optimizations in its network architecture and feature fusion mechanisms compared with earlier models such as YOLOv8 and YOLOv10. These improvements not only significantly enhance its detection performance for small-scale targets and complex scenarios but also ensure its leading position in terms of inference latency and model size. Experimental results on the MS COCO dataset demonstrate that YOLOv13-n achieves an mAP improvement of 3.0 and 1.5 percentage points over YOLOv11-n and YOLOv12-n, respectively.”
- While the model is described as “biomimetic,” the connection between primate visual mechanisms (e.g., foveation, gain control) and the technical improvements (DynamicConv, EMCA, Scale-based Loss) is somewhat superficial. The authors should more clearly articulate how each bio-inspired concept directly translates into the algorithmic design.
Modification instructions: We sincerely thank you for your insightful comments on the correspondence between biological inspiration and algorithmic design. We fully agree with this opinion and have revised the manuscript by adding detailed explanations for each improved module, elaborating the connection from primate visual mechanisms to algorithmic implementation, in order to address your concerns regarding superficial descriptions.
The supplementary content is as follows: “In primate vision, the receptive fields of the center and periphery modulate re-sponses in an antagonistic manner. Together with contrast gain control, this mechanism enables the visual system to dynamically adjust the effective receptive fields and channel responses according to stimulus intensity and background complexity. Inspired by this mechanism, the present study employs DynamicConv to optimize the DS-C3k2 modules and part of the convolutional layers in the Backbone and Neck, in order to simulate the dynamic regulation of effective receptive fields and channel gain in the visual system, thereby enhancing the model’s feature representation capability and detection robustness in complex tea garden scenarios.
In biological vision, information integration is achieved through two complemen-tary processes: global baseline estimation of brightness and texture for gain control, and selective enhancement of peak responses. Inspired by this mechanism, the present study introduces hybrid pooling into YOLOv13 to extract both global and peak statistics. A lightweight interaction is then constructed along the channel dimension to generate channel attention weights, which are reintegrated into the original features for channel recalibration. This process not only suppresses channel response overestimation caused by background noise but also highlights key responses related to object edges and textures, thereby improving the model’s discriminative capacity and robustness in complex tea garden scenarios. At the algorithmic level, this study builds upon the Efficient Channel Attention proposed by Qilong Wang et al. by introducing a joint strategy of Global Max Pooling and Global Average Pooling. The Efficient Mixed-pooling Channel Attention construct-ed after optimization is shown in Figure 5.
In biological vision, the fovea is responsible for high precision fixation, where small objects or task relevant details receive greater attention and resources, while salient and easily distinguishable large objects are encoded more coarsely. Correspondingly, in the loss design for object detection, this study adopts a similar principle by allowing the weights of the localization and scale terms to adapt dynamically to the actual object size. For small objects, the localization constraints are strengthened and the scale term is moderately relaxed, yielding more stable gradients and lower regression bias. For sufficiently salient large objects, the weights return to a more balanced configuration to ensure overall training stability. In this way, the biological allocation principle of finer encoding in the fovea and coarser representation in the periphery is translated into a scale aware dynamic reweighting strategy for the loss function, thereby improving the localization accuracy and convergence quality of small objects while maintaining balanced global performance. At the algorithmic level, this study optimizes the loss function of YOLOv13 by introducing Scale based Dynamic Loss without altering the original network architecture.”
- The ablation study results are presented in a dense table with many model variants, but the narrative does not sufficiently explain the contribution of each component. A clearer discussion of why certain combinations perform better—especially the interaction between modules—would strengthen the technical contribution.
Modification instructions: We sincerely appreciate your important suggestion. We fully agree that a clearer explanation of the independent contributions of each component and their interaction effects is necessary, and we have made the following substantive improvements in the revised manuscript.
“When the DynamicConv optimization is introduced separately, although the Parameters and Gradients are increased by 2.06 M, the FLOPs are reduced by 0.2 G, and the Precision, Recall, and mAP are increased by 1.7 %, 1.1 %, and 1.36 %, respectively. Experimental tests show that DynamicConv can expand the effective receptive field and adjust channel gain, thereby better representing fine-grained textures while reducing redundant static computations. With only Efficient Mixed-pooling Channel Attention, changes in parameters, gradients, and FLOPs remain within 1 percent, while precision, recall, and mAP increase by 2.74 percent, 0.09 percent, and 0.63 percent. The global gain estimation from GAP and GMP, combined with adaptive one-dimensional channel interaction, can to some extent suppress pseudo-responses such as glare and branch shadows, while selectively amplifying discriminative channels, thereby effectively improving precision. Scale-based Dynamic Loss keeps parameters, gradients, and FLOPs unchanged, yet raises precision, recall, and mAP by 0.92 percent, 1.72 percent, and 0.85 percent. The improved loss function can dynamically reweight the localization and scale terms according to object size, thereby alleviating regression bias for small objects and stabilizing gradient propagation. With the multi strategy configuration, parameters and gradients of DE-YOLOv13-S increase by 2.06M while computational complexity drops by 0.2G FLOPs, and precision, recall, and mAP rise by 3.78 percent, 2.04 percent, and 3.35 percent. The overall detection per-formance is significantly improved compared with the original network, which can be attributed to the complementary effects of the three proposed enhancements. DynamicConv effectively expands the effective receptive field and improves multi-scale representation, while Efficient Mixed-Channel Attention generates more robust channel attention weights. In addition, the dynamic adjustment of localization constraints for small objects in Scale-based Dynamic Loss Optimization further enhances the final model’s localization accuracy for small targets.”
- The paper claims practical utility for tea garden management but does not discuss deployment constraints (e.g., computational cost on edge devices, inference speed). A discussion on real-time performance or scalability would enhance the practical relevance of the work.
Modification instructions: We sincerely appreciate your constructive suggestion regarding practical deployment constraints and real time performance. We fully agree that a clearer discussion on computational cost and inference speed on devices is necessary. Therefore, in addition to the original ablation experiments that systematically compared FLOPs, Parameters, and Gradients, we have added the FPS metric to Table 2 and the ablation study discussion to more explicitly address your concern on real time performance.
The supplementary content is as follows: “In terms of detection speed, DynamicConv optimization benefits from the reduction in FLOPs and improves the FPS by 2.40 compared with the original network. The scale adaptive constraints introduced by Scale based Dynamic Loss enhance the regression stability of small objects, indirectly reducing computational overhead and increasing the FPS by 1.58 over the original network. In contrast, Efficient Mixed pooling Channel Attention introduces a small amount of channel interaction, resulting in a slight decrease of 0.38 FPS. After integrating all improvements, the overall FPS of the model shows a net increase of 1.99, indicating a notable improvement in detection speed.”
- The manuscript should be carefully reviewed to ensure all external sources are properly cited and that the text presents original synthesis and phrasing. While some similarity is expected in technical writing due to standard terminology and methods, the authors must verify that all borrowed ideas and text are appropriately referenced to maintain academic integrity. A thorough check for correct paraphrasing and quotation is recommended before publication.
Modification instructions: We sincerely thank you for your careful review and valuable suggestions. We have thoroughly checked and revised the manuscript section by section. For paragraphs involving the views, data, or methodological descriptions of others, we have supplemented and corrected the relevant references to ensure academic integrity and consistency with original expressions. To make the article more concise, we have provided citations only at the first occurrence of related content; however, if you prefer citations to appear at every instance, please let us know and we will promptly add and adjust them. Should there still be any omissions or shortcomings, we kindly invite further feedback, and we will make the necessary improvements without delay. Thank you very much.
- The manuscript is generally well-written, with clear technical descriptions and coherent structure. However, there are occasional grammatical inconsistencies, awkward phrasings, and punctuation issues (e.g., missing articles, inconsistent tense usage). A thorough proofreading by a native English speaker is recommended to improve readability and polish.
Modification instructions: We sincerely appreciate your careful suggestion. We have thoroughly checked and revised the manuscript sentence by sentence, with a particular focus on unifying tense and voice, completing and standardizing the use of articles, eliminating awkward expressions and redundant wording, and comprehensively checking and correcting inconsistencies in punctuation and formatting, in order to further enhance the readability of the text.

Reviewer 3 Report
Comments and Suggestions for Authors
- Please provide the meanings of the horizontal and vertical coordinates for each subgraph in Figures 7 and 8.
- Please provide the detailed data distribution of the training set, validation set, and testing set used in this article in a clear table format.
- The introduction section briefly explains why primates are chosen for their visual mechanisms. Please provide more theoretical support in the field of bionics.
- Please supplement the performance of the proposed detection model compared to other models in terms of single frame image detection time (fps) metric. And further compare and analyze the performance differences in detection speed among various models.
- Tables 2 and 3 should be supplemented with quantitative data on detection indicators for target substances in categories A, B, and C, and comparative analysis should be conducted.
- What are the specific limitations of this study? Please have the author summarize and condense several points to explain and clarify.
Author Response
Thanks very much for your time to review this manuscript. I really appreciate you’re your comments and suggestions. We have considered these comments carefully and tried our best to address every one of them. And the corresponding part in the text has been modified using red font.
- Please provide the meanings of the horizontal and vertical coordinates for each subgraph in Figures 7 and 8.
Modification instructions: We sincerely appreciate your valuable suggestion. We have added clarifications in the manuscript regarding the coordinate meanings of Figures 7 and 8. In Figure 7, the X axis represents the number of training epochs, and the Y axis denotes the loss value. In Figure 8, the X axis in all subfigures represents the confidence threshold. The Y axis of Figures 8A and 8D corresponds to Precision, Figures 8B and 8E to Recall, and Figures 8C and 8F to F1.
- Please provide the detailed data distribution of the training set, validation set, and testing set used in this article in a clear table format.
Modification instructions: We sincerely appreciate your valuable suggestion. We have added clear tables and descriptions in the manuscript to present the division and quantities of the dataset. In this study, a total of 2,147 original images and 7,403 object labels were annotated. The original images were divided into training and test sets in an 8:2 ratio, with 80% used for five fold cross validation training and 20% reserved for final model testing. In addition, we supplemented an independent external validation set consisting of 500 images and 1,864 labels, which has also been included in the table for ease of review.
- The introduction section briefly explains why primates are chosen for their visual mechanisms. Please provide more theoretical support in the field of bionics.
Modification instructions: We sincerely thank you for your insightful comments on the correspondence between biological inspiration and algorithmic design. We fully agree with this opinion and have revised the manuscript by adding detailed explanations for each improved module, elaborating the connection from primate visual mechanisms to algorithmic implementation.
The supplementary content is as follows: “In primate vision, the receptive fields of the center and periphery modulate re-sponses in an antagonistic manner. Together with contrast gain control, this mechanism enables the visual system to dynamically adjust the effective receptive fields and channel responses according to stimulus intensity and background complexity. Inspired by this mechanism, the present study employs DynamicConv to optimize the DS-C3k2 modules and part of the convolutional layers in the Backbone and Neck, in order to simulate the dynamic regulation of effective receptive fields and channel gain in the visual system, thereby enhancing the model’s feature representation capability and detection robustness in complex tea garden scenarios.
In biological vision, information integration is achieved through two complemen-tary processes: global baseline estimation of brightness and texture for gain control, and selective enhancement of peak responses. Inspired by this mechanism, the present study introduces hybrid pooling into YOLOv13 to extract both global and peak statistics. A lightweight interaction is then constructed along the channel dimension to generate channel attention weights, which are reintegrated into the original features for channel recalibration. This process not only suppresses channel response overestimation caused by background noise but also highlights key responses related to object edges and textures, thereby improving the model’s discriminative capacity and robustness in complex tea garden scenarios. At the algorithmic level, this study builds upon the Efficient Channel Attention proposed by Qilong Wang et al. by introducing a joint strategy of Global Max Pooling and Global Average Pooling. The Efficient Mixed-pooling Channel Attention construct-ed after optimization is shown in Figure 5.
In biological vision, the fovea is responsible for high precision fixation, where small objects or task relevant details receive greater attention and resources, while salient and easily distinguishable large objects are encoded more coarsely. Correspondingly, in the loss design for object detection, this study adopts a similar principle by allowing the weights of the localization and scale terms to adapt dynamically to the actual object size. For small objects, the localization constraints are strengthened and the scale term is moderately relaxed, yielding more stable gradients and lower regression bias. For sufficiently salient large objects, the weights return to a more balanced configuration to ensure overall training stability. In this way, the biological allocation principle of finer encoding in the fovea and coarser representation in the periphery is translated into a scale aware dynamic reweighting strategy for the loss function, thereby improving the localization accuracy and convergence quality of small objects while maintaining balanced global performance. At the algorithmic level, this study optimizes the loss function of YOLOv13 by introducing Scale based Dynamic Loss without altering the original network architecture.”
- Please supplement the performance of the proposed detection model compared to other models in terms of single frame image detection time (fps) metric. And further compare and analyze the performance differences in detection speed among various models.
Modification instructions: We sincerely appreciate your constructive suggestion regarding practical deployment constraints and real time performance. We fully agree that a clearer discussion on computational cost and inference speed on devices is necessary. Therefore, in addition to the original ablation experiments that systematically compared FLOPs, Parameters, and Gradients, we have added the FPS metric to Table 2 and the ablation study discussion to more explicitly address your concern on real time performance.
The supplementary content is as follows: “In terms of detection speed, DynamicConv optimization benefits from the reduction in FLOPs and improves the FPS by 2.40 compared with the original network. The scale adaptive constraints introduced by Scale based Dynamic Loss enhance the regression stability of small objects, indirectly reducing computational overhead and increasing the FPS by 1.58 over the original network. In contrast, Efficient Mixed pooling Channel Attention introduces a small amount of channel interaction, resulting in a slight decrease of 0.38 FPS. After integrating all improvements, the overall FPS of the model shows a net increase of 1.99, indicating a notable improvement in detection speed.”
- Tables 2 and 3 should be supplemented with quantitative data on detection indicators for target substances in categories A, B, and C, and comparative analysis should be conducted.
Modification instructions: We sincerely appreciate your constructive comments. We have supplemented the manuscript with quantitative results and comparative analyses of the detection metrics for the three target categories A, B, and C. Compared with YOLOv13, SSD, CornerNet, and RT-DETR, the improved DE-YOLOv13-S achieves AP improvements of 2.59%, 13.56%, 16.44%, and 5.35% for tender bud recognition, 3.08%, 14.79%, 16.29%, and 5.39% for one bud and one leaf recognition, and 4.38%, 10.89%, 13.50%, and 4.78% for one bud and two leaves recognition. Since Table 3 (formerly Table 2) has reached its maximum length, we have included the newly added quantitative results in Table 4 (formerly Table 3) and provided the corresponding comparative discussion in the text. If further expansion of the data or adjustments in presentation are required, please feel free to let us know, and we will either supplement the results or present them in a separate new table.
- What are the specific limitations of this study? Please have the author summarize and condense several points to explain and clarify.
Modification instructions: We sincerely appreciate your suggestion, and we have further supplemented the main limitations of this study in the revised manuscript. It should be noted that although this study includes an external validation set, the data collection sites remain relatively limited, and the generalizability to other regions, tea varieties, and extreme climatic conditions still requires further validation. Moreover, while the improved network demonstrates notable improvements under dense occlusion, strong reflections, and extreme illumination, its robustness still shows a certain degree of decline. In the future, the research team will further explore multimodal sensing data fusion, cross regional model transferability, and few shot learning strategies, while continuously expanding the scope of research regions and objects to broaden the applicability and adaptability of this approach in intelligent detection of various economic crops. In addition, we plan to introduce strategies such as image augmentation to perform additional processing on the input images, so as to enhance the practicality of the model under extreme illumination and other challenging environmental conditions.

Reviewer 4 Report
Comments and Suggestions for Authors
The topic is important, and the proposed modification of YOLOv13 appears interesting, with compelling ideas. The results are presented as a substantial improvement over the up-to-date YOLOv13 model.
The primary question that arises concerns the testing protocol. The authors report results on the validation set and do not mention a test set. However, if we understand the validation set in the standard way, it is used for hyperparameter tuning and model selection, while the test set is held out and never used in any form of parameter selection. Therefore, the significance of the reported results is unclear. It raises concerns about potential overfitting or data leakage. Please clarify this situation. If a separate test set was not used, we strongly recommend creating one and providing the results on it. If it was used, please detail how the data split was performed.
Furthermore, the authors state: “The results show that the Box Loss, Cls Loss, and DFL Loss of the DE-YOLOv13-S network decreased by 18.75%, 3.70%, and 2.54% on the training set, and by 18.48%, 14.29%, and 7.46% on the validation set, respectively. Compared with YOLOv13, <…> Precision, Recall and mAP are increased by 3.78 %, 2.04 % and 3.35 %, respectively.”
On which dataset were these final Precision, Recall, and mAP metrics calculated? How can we explain the large advantage in the loss reduction versus the relatively modest improvement in the final detection metrics? We note that a 3% improvement on a proper test set would indeed be very substantial. The observed discrepancy requires explanation.
Author Response
Thanks very much for your time to review this manuscript. I really appreciate you’re your comments and suggestions. We have considered these comments carefully and tried our best to address every one of them. And the corresponding part in the text has been modified using red font.
- The primary question that arises concerns the testing protocol. The authors report results on the validation set and do not mention a test set. However, if we understand the validation set in the standard way, it is used for hyperparameter tuning and model selection, while the test set is held out and never used in any form of parameter selection. Therefore, the significance of the reported results is unclear. It raises concerns about potential overfitting or data leakage. Please clarify this situation. If a separate test set was not used, we strongly recommend creating one and providing the results on it. If it was used, please detail how the data split was performed.
Modification instructions: We sincerely appreciate your valuable suggestion. We have added clear tables and descriptions in the manuscript to present the division and quantities of the dataset. In this study, a total of 2,147 original images and 7,403 object labels were annotated. The original images were divided into training and test sets in an 8:2 ratio, with 80% used for five fold cross validation training and 20% reserved for final model testing. In addition, we supplemented an independent external validation set consisting of 500 images and 1,864 labels, which has also been included in the table for ease of review.
- Furthermore, the authors state: “The results show that the Box Loss, Cls Loss, and DFL Loss of the DE-YOLOv13-S network decreased by 18.75%, 3.70%, and 2.54% on the training set, and by 18.48%, 14.29%, and 7.46% on the validation set, respectively. Compared with YOLOv13, <…> Precision, Recall and mAP are increased by 3.78 %, 2.04 % and 3.35 %, respectively.” On which dataset were these final Precision, Recall, and mAP metrics calculated?
Modification instructions: We sincerely appreciate your correction and inquiry. The Precision, Recall, and mAP reported in the manuscript are all calculated based on the final test set. This test set was drawn from the 20% reserved portion of the original dataset and is entirely independent of the dataset used for five fold cross validation, without participating in any training, hyperparameter tuning, or model selection. After completing the five fold cross validation during training, we further evaluated the independent test set once, from which the above metrics were obtained. We have already provided a detailed description and correction of this part in the revised manuscript, and we sincerely thank you again for your suggestion.
- How can we explain the large advantage in the loss reduction versus the relatively modest improvement in the final detection metrics? We note that a 3% improvement on a proper test set would indeed be very substantial. The observed discrepancy requires explanation.
Modification instructions: We sincerely appreciate your insightful suggestion. It should be noted that although the percentage reduction in loss appears large, the original loss magnitude was already around 1, so the absolute decrease is only relatively evident rather than showing a substantial advantage. In addition, the reductions in Box, Cls, and DFL primarily reflect improvements in bounding box regression stability and confidence distribution calibration, while the final mAP remains influenced by multiple factors such as IoU distribution and class imbalance. When the model operates in a higher-accuracy regime, the evaluation metrics exhibit diminishing marginal returns, which explains why a relatively large decrease in loss may correspond to only moderate improvements in performance metrics.
In addition, we sincerely appreciate your positive recognition of our research results. In response to this comment, we have reorganized the ablation experiments to present more clearly the independent contributions of each component as well as the sources and cumulative effects of their interactions on overall performance. Furthermore, we have introduced FPS as an additional metric to investigate the inference speed of the model in practical operation, thereby reflecting its overall effectiveness in real deployment scenarios.
The supplementary content is as follows: “When the DynamicConv optimization is introduced separately, although the Parameters and Gradients are increased by 2.06 M, the FLOPs are reduced by 0.2 G, and the Precision, Recall, and mAP are increased by 1.7 %, 1.1 %, and 1.36 %, respectively. Experimental tests show that DynamicConv can expand the effective receptive field and adjust channel gain, thereby better representing fine-grained textures while reducing redundant static computations. With only Efficient Mixed-pooling Channel Attention, changes in parameters, gradients, and FLOPs remain within 1 percent, while precision, recall, and mAP increase by 2.74 percent, 0.09 percent, and 0.63 percent. The global gain estimation from GAP and GMP, combined with adaptive one-dimensional channel interaction, can to some extent suppress pseudo-responses such as glare and branch shadows, while selectively amplifying discriminative channels, thereby effectively improving precision. Scale-based Dynamic Loss keeps parameters, gradients, and FLOPs unchanged, yet raises precision, recall, and mAP by 0.92 percent, 1.72 percent, and 0.85 percent. The improved loss function can dynamically reweight the localization and scale terms according to object size, thereby alleviating regression bias for small objects and stabilizing gradient propagation. With the multi strategy configuration, parameters and gradients of DE-YOLOv13-S increase by 2.06M while computational complexity drops by 0.2G FLOPs, and precision, recall, and mAP rise by 3.78 percent, 2.04 percent, and 3.35 percent. The overall detection per-formance is significantly improved compared with the original network, which can be attributed to the complementary effects of the three proposed enhancements. DynamicConv effectively expands the effective receptive field and improves multi-scale representation, while Efficient Mixed-Channel Attention generates more robust channel attention weights. In addition, the dynamic adjustment of localization constraints for small objects in Scale-based Dynamic Loss Optimization further enhances the final model’s localization accuracy for small targets. In terms of detection speed, DynamicConv optimization benefits from the reduction in FLOPs and improves the FPS by 2.40 compared with the original network. The scale adaptive constraints introduced by Scale based Dynamic Loss enhance the regression stability of small objects, indirectly reducing computational overhead and increasing the FPS by 1.58 over the original network. In contrast, Efficient Mixed pooling Channel Attention introduces a small amount of channel interaction, resulting in a slight decrease of 0.38 FPS. After integrating all improvements, the overall FPS of the model shows a net increase of 1.99, indicating a notable improvement in detection speed.”

Round 2
Reviewer 1 Report
Comments and Suggestions for Authors
The comments are addressed
Author Response
Thank you very much for your positive feedback and affirmation. We truly appreciate your recognition of our work and the time you devoted to reviewing our manuscript.
Reviewer 2 Report
Comments and Suggestions for Authors
The authors have thoroughly revised the manuscript for the reviewer comments.
Author Response

(The authors gave the same response as above.)

Reviewer 4 Report
Comments and Suggestions for Authors
The authors have provided full answers to all the questions and comments.
The only issue is that the abstract does not mention what improvements were achieved on the test dataset.
Author Response
Thanks very much for your time to review this manuscript. I really appreciate you’re your comments and suggestions. We have considered these comments carefully and tried our best to address every one of them. And the corresponding part in the text has been modified using red font.
- The authors have provided full answers to all the questions and comments.
Modification instructions: Thank you very much for your positive feedback and affirmation. We truly appreciate your recognition of our work and the time you devoted to reviewing our manuscript.
- The only issue is that the abstract does not mention what improvements were achieved on the test dataset.
Modification instructions: Thank you very much for your valuable comment. We sincerely appreciate your careful review and constructive suggestion. In response, we have revised the abstract to include the improvements achieved on the test dataset, and have also updated the corresponding expressions in the main text to ensure consistency and clarity.